# Detection of Thrombosis Using Soluble C-Type Lectin-like Receptor-2 with D-Dimer Level and Platelet Count

**DOI:** 10.3390/jcm13195980

**Published:** 2024-10-08

**Authors:** Hideo Wada, Katsuya Shiraki, Akitaka Yamamoto, Toshitaka Kamon, Jun Masuda, Yuhuko Ichikawa, Masahide Kawamura, Motomu Shimaoka, Hideto Simpo

**Affiliations:** 1Department of General and Laboratory Medicine, Mie Prefectural General Medical Center, Yokkaichi 510-8561, Japan; katsuya-shiraki@mie-gmc.jp; 2Department of Emergency and Critical Care Center, Mie Prefectural General Medical Center, Yokkaichi 510-8561, Japan; akitaka-yamamoto@mie-gmc.jp; 3Department of Neurology, Mie Prefectural General Medical Center, Yokkaichi 510-8561, Japan; toshi9031@yahoo.co.jp; 4Department of Cardiovascular Medicine, Mie Prefectural General Medical Center, Yokkaichi 510-8561, Japan; jun-masuda@mie-gmc.jp; 5Department of Central Laboratory, Mie Prefectural General Medical Center, Yokkaichi 510-8561, Japan; ichi911239@yahoo.co.jp; 6In Vitro Diagnostic Division, PHC Corporation, Tokyo 174-0053, Japan; masahide.kawamura@phchd.com; 7Department of Molecular Pathobiology and Cell Adhesion Biology, Mie University Graduate School of Medicine, Tsu 514-8507, Japan; motomushimaoka@gmail.com; 8Mie Prefectural General Medical Center, Yokkaichi 510-8561, Japan; hideto-shimpo@mie-gmc.jp

**Keywords:** sCLEC-2, micro thrombosis, arterial thrombosis, venous thrombosis, D-dimer, platelet count

## Abstract

**Introduction**: Soluble C-type lectin-like receptor -2 (sCLEC-2) has been recognized as a marker of platelet activation, and attention has been drawn to formulas combining sCLEC-2 levels with platelet count and D-dimer levels. **Methods**: In this study, sCLEC-2 levels, as well as sCLEC-2/platelet count (sCLEC-2/PLT), sCLEC-2 × D-dimer (sCLEC-2xDD), and sCLEc-2xDD/PLT formulas were used to detect thrombotic diseases, including microvascular thrombosis (MVT), arterial thromboembolism (ATE), and venous thromboembolism (VTE), with the aim of evaluating the ability of the three parameters combined in these formulas to diagnose thrombotic diseases. **Results**: The plasma sCLEC-2 levels were significantly higher in patients with infectious or thrombotic diseases than in those with neither thrombosis nor infection; however, there was no significant difference among patients with infection, ATE, VTE, and MVT; the correlations among sCLEC-2, platelet count, and D-dimer level were poor. The sCLEC-2/PLT ratio was the highest in patients with MVT, and the sCLEC-2 × D-dimer value was higher in patients with MVT and VTE than in those with neither thrombosis nor infection. Although receiver operating characteristic (ROC) analysis shows the differential diagnosis of thrombotic diseases from non-thrombosis without infection, the sCLEC-2 × D-dimer/platelet count was useful for differential diagnosis among MVT and infection or non-thrombotic diseases. **Conclusions**: sCLEC-2 is useful for the diagnosis of thrombosis, and the formulas of sCLEC-2 with platelet count or D-dimer are useful for the diagnosis of thrombosis using ROC analyses for the thrombosis group vs. the non-thrombosis group without infection.

## 1. Introduction

Thrombosis is classified into venous thromboembolism (VTE) [1,2,3], such as deep vein thrombosis and pulmonary embolism, and arterial thromboembolism (ATE), such as cerebral infarction [4] and acute myocardial infarction (AMI) [5]. ATE, which causes ischemic symptoms of the affected organ, is considered to be caused mainly by platelet activation [6]; on the other hand, VTE, which causes various symptoms from edema to ischemia, is mainly caused by the activation of the coagulation system [7]. Therefore, VTE is generally treated with anticoagulants [1,2], while ATE is generally treated with antiplatelet therapy [8]; thus, differential diagnosis between ATE and VTE is important, and their early diagnosis and treatment are vital, given the sudden onset and potentially fatal result of thrombosis.

There are a few biomarkers of platelet activation [9,10] that are useful for clinical applications in the differential diagnosis of thrombocytopenia [11,12,13], or in monitoring antiplatelet therapy for arterial thrombosis [14,15]. β-Thromboglobulin (β-TG) [16], P-selectin [17], and platelet factor 4 (PF4) [18] have been previously reported as biomarkers of platelet activation; however, their actual specificity for the diagnosis of arterial thrombosis and the evaluation of antiplatelet therapy efficacy are not high, and they have limited use in the clinical laboratory setting. Flow cytometry can be used to detect P-selectin [19] or phosphatidylserine [20] on the platelet membrane in order to evaluate platelet activation, but it is not a routine laboratory examination. Although soluble platelet glycoprotein VI has also been reported to be useful for detecting platelet activation [21,22], this assay is not available for clinical use. 

C-type lectin-like receptor 2 (CLEC-2) is a receptor for platelet-activating snake venom, rhodocytin/aggretin, and it is almost specifically expressed in platelets/megakaryocytes in humans [23,24]; measurement of the soluble form of CLEC-2 (sCLEC-2) has been developed as a new biomarker for platelet activation [25,26], and this assay is not affected by the blood sampling procedure or centrifugation. The measurement of sCLEC-2 has been reported to be useful for the diagnosis of microthrombosis such as disseminated intravascular coagulation (DIC) [27] and thrombotic microangiopathy (TMA) [28]; macrothrombosis, such as acute cerebral infarction (ACI) [29,30] and AMI [31,32]; and infectious diseases, such as COVID-19 [33,34,35]. 

An elevated D-dimer level has also been reported in DIC [36,37], VTE [38,39], ACI [40], and AMI [41]. In addition, thrombocytopenia has been reported in microvascular thrombosis (MVT) including DIC [42] and TMA [43]. A scoring system, including fibrin-related marker, platelet count, prothrombin time, and fibrinogen, has been used to diagnose DIC [44]; additionally, the super formulas sCLEC-2 × D-dimer/platelet count (sCLEC-2xDD/PLT) and sCLEC-2 × D-dimer (sCLEC-2xDD) were recently reported to increase diagnostic ability for DIC and ACI, respectively [45,46]. Furthermore, the sCLEC-2/D-dimer (sCLEC-2/DD) and sCLEC-2/platelet count (sCLEC-2/PLT) ratios have been reported to be useful for the diagnosis of DIC [47], as well as for the differential diagnosis of atherosclerotic ACI from cardioembolic ACI [30].

In the present study, we examined plasma sCLEC-2 and D-dimer levels as well as platelet counts in 330 patients with neither thrombosis nor infection, 141 patients with infection, and 204 patients with thrombotic diseases, including 90 patients with MVT, 92 with ATE, and 22 with VTE. We evaluated formulas combining these three parameters for the diagnosis of thrombotic diseases.

## 2. Materials and Methods

Plasma sCLEC-2 levels, platelet counts, and D-dimer levels were measured in 79 healthy volunteers (HVs); 52 patients with indefinite compliant syndrome (ICS); 35 samples from 22 patients with immune thrombocytopenia (ITP); 177 patients with chronic liver disease (CLD); 141 infection (IFN) patients without thrombosis; 90 patients with MVT, including 32 patients with TMA (thrombotic thrombocytopenic purpura, 20 patients; secondary TMA, 12 patients), and 58 patients with DIC; 92 patients with ATE, including 72 ACI patients without atrial fibrillation; 20 patients with AMI; and 36 samples from 22 patients with VTE. All measurements were performed at Mie Prefectural General Medical Center from 1 September 2019 to 31 March 2024 (Table 1). ICS is used to describe a condition in which patients experience a variety of vague, non-specific symptoms such as fatigue, dizziness, and general discomfort but medical examinations and tests do not reveal any specific abnormalities [48]. The CLD group was not associated with thrombosis or thrombocytopenia and included 96 patients with chronic hepatitis, 51 with fatty liver disease, 15 with alcoholic liver damage, and 15 with autoimmune hepatitis. Patients with liver cirrhosis or hepatic cell carcinoma were excluded from this group. DIC was diagnosed using the ISTH overt DIC diagnostic criteria [49]. Thrombotic and infectious events were not observed in HVs, ICS, ITP, or CLD, and these conditions were considered to represent a state with neither thrombosis nor bacterial infection (TH-, IFN-). IFN patients did not include those with severe bacterial infection and were not associated with DIC or thrombosis. This study’s protocol (O-0051) was approved by the Human Ethics Review Committee of Mie Prefectural General Medical Center, and informed consent was obtained from each participant. This study was conducted in accordance with the principles of the Declaration of Helsinki. 

Blood sampling was performed in the early morning, during bed rest, from day 1 to day 10 from admission. Plasma was prepared in two centrifugations at 3000 rpm for 15 min (platelet count < 0.5 × 10^10^/L). Plasma sCLEC-2 levels were measured with a chemiluminescent enzyme immunoassay (CLEIA) using previously described monoclonal antibodies and the STACIA CLEIA system (PHC Holdings Co., Tokyo, Japan) [25,26,27]. The sensitivity (limit of quantification) of the sCLEC-2 CLEIA method is 14.1 pg/mL (according to the package insert of the kit), allowing for the measurement of the levels in samples from both healthy individuals and patients. The specificity of the antibodies used in the CLEIA reagents is described in detail in the analyses conducted using the ELISA and Western blot methods [31,50]; these antibodies recognize both the 25 kD (shed form) and the 32, 40 kD (microparticle form) variants of sCLEC-2. Plasma D-dimer levels were measured using LPIA-Genesis (PHC Holdings Co.) with the STACIA system (PHC Holdings Co.). Platelet counts were measured using a fully automatic blood cell counter (XN-3000, Sysmex Co. Kobe, Japan).

### Statistical Analyses

Data are expressed as median values (25th–75th percentiles). The significance of the differences between groups was examined using the Mann–Whitney *U*-test. The cutoff values, areas under the curve (AUCs), sensitivity, specificity, and odds ratios were determined using receiver operating characteristic (ROC) analyses. Statistical significance was set at *p* < 0.05. All statistical analyses were performed using the Stat-Flex software program (version 7; Artec Co Ltd., Osaka, Japan).

## 3. Results

Plasma sCLEC-2 levels were significantly lower in patients with ITP than in HVs, but significantly higher in patients with ICS, CLD, IFN, MVT, ATE, or VTE in comparison to HVs (Figure 1A and Table 1). Plasma sCLEC-2 levels were significantly higher in patients with IFN, or thrombotic diseases, including MVT, ATE, and VTE, than in those with neither thrombotic diseases nor IFN, including those with ITP, ICS, or CLD. Plasma sCLEC-2 levels among patients with thrombotic diseases and IFN were highest in those with MVT; however, there were no significant differences in plasma sCLEC-2 levels between the IFN and ATE or VTE groups. Plasma D-dimer levels were significantly higher in patients with thrombotic diseases or IFN than in those with neither thrombotic diseases nor IFN (Figure 1B and Table 1). Platelet counts were significantly lower in patients with ITP or MVT than in those with ICS, CLD, IFN, ATE, or VTE (Figure 1C and Table 1). Correlations between sCLEC-2 levels and platelet counts (r = 0.079) and between sCLEC-2 and D-dimer levels (r = 0.236) were poor (Figure 2). Regarding ROC analyses of sCLEC-2 for the diagnosis of thrombosis vs. for patients with neither thrombosis nor infection (Table 2), the AUCs were 0.977 for MVT, 0.979 for ATE and 0.967 for VTE. However, ROC analyses of sCLEC-2 for the diagnosis of thrombosis vs. for patients with infection showed that the AUCs were less than 0.65 for MVT, ATE, or VTE. The sCLEC-2/platelet count ratio among patients with neither thrombotic diseases nor IFN was the highest in those with ITP, and among patients with IFN and thrombotic diseases, was the highest in those with MVT (Figure 3A and Table 3).

The sCLEC-2 × D-dimer value was higher in patients with thrombotic diseases than in those with neither thrombotic diseases nor IFN, and the sCLEC-2 × D-dimer value was the highest in patients with MVT, among all diseases (Figure 3B and Table 3). The sCLEC-2 × D-dimer/platelet count value was significantly higher in patients with thrombotic diseases than in those with neither thrombotic diseases nor IFN, and the sCLEC-2 × D-dimer/platelet count value was the highest in patients with MVT, among all diseases (Figure 3C). Regarding ROC analyses of the sCLEC-2/platelet count, sCLEC-2 × D-dimer, and sCLEC-2 × D-dimer/platelet count formulas for the diagnosis of thrombosis vs. patients with neither thrombosis nor infection (Table 4), the AUCs were 0.988, 0.998, and 0.999, respectively, in MVT; 0.899, 0.971, and 0.934, respectively, in ATE; and 0.930, 0.983, and 0.965, respectively, in VTE. The ROC analyses of the sCLEC-2/platelet count, sCLEC-2 × D-dimer, and sCLEC-2 × D-dimer/platelet count formulas for the diagnosis of thrombosis vs. for patients with infection showed that the AUCs were 0.908, 0.839, and 0.941, respectively, in MVT; 0.559, 0.717, and 0.725, respectively, in ATE; and 0.557, 0.680, and 0.660, respectively, in VTE.

## 4. Discussion

Increased sCLEC-2 levels, which are considered to be biomarkers of platelet activation [51,52], are present in most underlying diseases except ITP, which suggests that platelet activation occurs in these diseases. Although the plasma sCLEC-2 levels were significantly lower in patients with ITP, which is frequently associated with severe thrombocytopenia [12,53], than in HVs, and patients with ICS or CLD, the sCLEC-2/PLT ratio was significantly higher in ITP than HVs or these other patients, suggesting that platelets might be mildly activated in ITP, or that there is a release of sCLEC-2 from platelets phagocytosed by splenic macrophages [54]. It has been reported that ITP was sometimes clinically associated with thrombosis [55,56]; although our findings showed a poor correlation between plasma sCLEC-2 levels and platelet count, sCLEC-2 elevation may not be significant in thrombocytopenia due to ITP. Therefore, the usefulness of the sCLEC-2/PLT ratio was reported in DIC [45,46]; in the present study, the ratio was markedly high in both TMA and DIC, suggesting that it can be useful for the differential diagnosis of thrombocytopenia, such as aplastic anemia [57], ITP [58], and TMA [59]. 

Markedly increased sCLEC-2 levels indicate that platelet activation occurs in infectious and thrombotic diseases, additionally suggesting that infectious diseases are often associated with thrombosis. It was reported that platelets were stimulated by COVID-19 infection, depending on the CD-147 receptor [36]; plasma sCLEC-2 levels were also significantly higher in COVID-19 patients than in bacterial infection patients and patients from previous reports [33,34]. Our findings also suggest that elevated sCLEC-2 levels are useful for the diagnosis of thrombosis in patients without infection. D-dimer is a biomarker for the activation of the coagulation system and can predict and negatively exclude VTE [39,60]; therefore, the formula, sCLEC-2xDD is considered a biomarker for the activation of both platelets and the coagulation system, and our findings showed that this formula was the highest in MVT and the second highest in VTE, suggesting that these diseases are markedly hypercoagulable and require treatment with anticoagulants such as heparin, direct oral anticoagulants, warfarin, or antiplatelet therapy. 

Atherosclerotic cerebral infarction should be treated with antiplatelet therapy [61,62], whereas cardiogenic cerebral infarction should be treated with direct oral anticoagulants or warfarin [63,64]; however, few biomarkers are available for the prediction of atherosclerotic cerebral infarction. Our findings confirm that the formula of sCLEC-2xDD may predict thrombotic diseases, including atherosclerotic and cardiogenic cerebral infarction, and other sCLEC-2/DD formulas may differentially diagnose atherosclerotic cerebral infarction from cardiogenic cerebral infarction. 

MVT showed the highest sCLEC-2xDD/PLT ratio among all diseases, suggesting that this formula is the most useful for the diagnosis of microvascular thrombosis, as previously reported [45]. ROC analyses showed that the sCLEC-2 × D-dimer/platelet count formula was better than the sCLEC-2 × D-dimer formula for the diagnosis of thrombosis in patients with infections, but not in patients without infection. 

A summarized figure (Figure 4) shows the relationships between formulas with sCLEC-2 levels, D-dimer levels, and platelet counts, and thrombosis including ATE, MVT, and VTE. sCLEC-2 levels were high in all thrombosis types, with sCLEC-2 in descending order of MVT, ATE, and VTE, and D-dimer levels in descending order of VTE, MVT, and ATE. Platelet counts were low in only MVT. The sCLEC-2 × D-dimer was high in ATE and VTE, and especially high in MVT. sCLEC-2/D-dimer ratios were high in ATE and low in VTE. sCLEC-2 x platelet count values were high in ATE and VTE but low in MVT. The sCLEC-2/platelet count and sCLEC-2 × D-dimer/platelet count ratios were high in ATE and VTE, and markedly high in MVT. These findings suggest that sCLEC-2 is useful for the diagnosis of ATE and MVT, but D-dimer levels are useful for the diagnosis of VTE and MVT. Therefore, the sCLEC-2 × D-dimer formula is useful for the diagnosis of thrombosis. The sCLEC-2/D-dimer ratio is useful for the differential diagnosis between ATE and VTE, and the sCLEC-2/PLT and sCLEC-2 × D-dimer/PLT ratios are most useful for the diagnosis of MVT.

## 5. Conclusions

The sCLEC-2 and sCLEC-2 × D-dimer formulas are useful for the diagnosis of thrombosis in patients without infection using ROC analyses. The sCLEC-2 × D-dimer, sCLEC-2/platelet count, and sCLEC-2 × D-dimer/platelet count may increase the ability to predict thrombosis. The sCLEC-2/platelet count and sCLEC-2 × D-dimer/platelet count formulas are useful for predicting thrombosis in patients without infection, and the sCLEC-2xDD/PLT formula is also highly useful for predicting MVT.

## Figures and Tables

**Figure 1 jcm-13-05980-f001:**
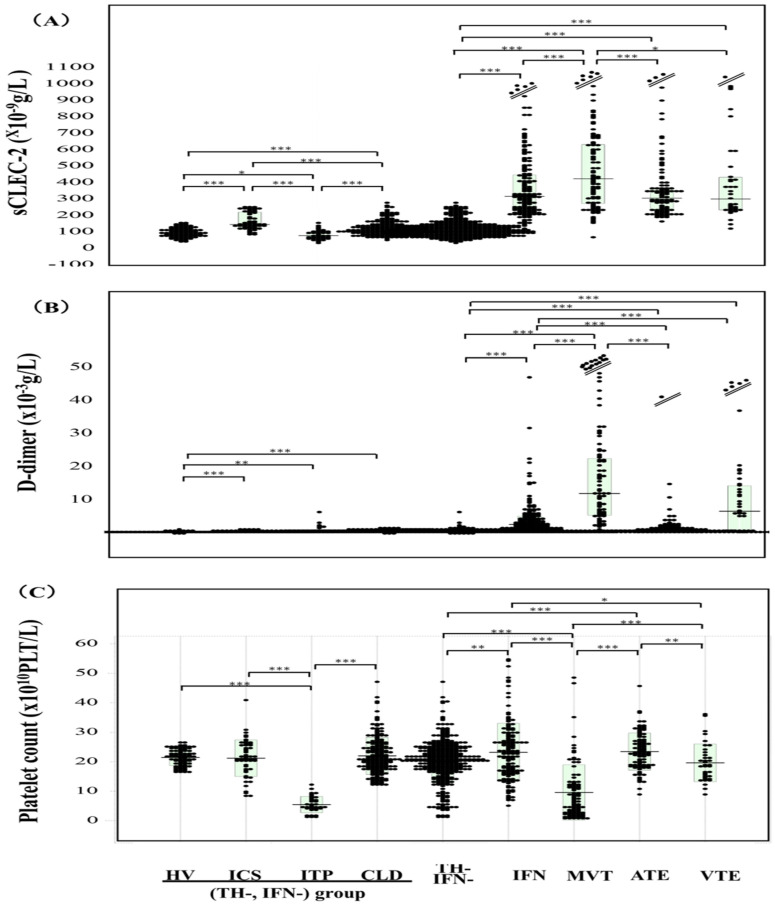
Plasma sCLEC-2 levels (**A**), D-dimer levels (**B**), and platelet counts (**C**) in HVs and patients with various diseases. HV, healthy volunteer; ICS, indefinite compliant syndrome; ITP, immune thrombocytopenia; CLD, chronic liver disease; INF, infection; TH-, without thrombosis; IFN-, without infection; MVT; microvascular thrombosis; VTE, venous thromboembolism; ATE, arterial thromboembolism; *, *p* < 0.05; **, *p* < 0.01; ***, *p* < 0.001.

**Figure 2 jcm-13-05980-f002:**
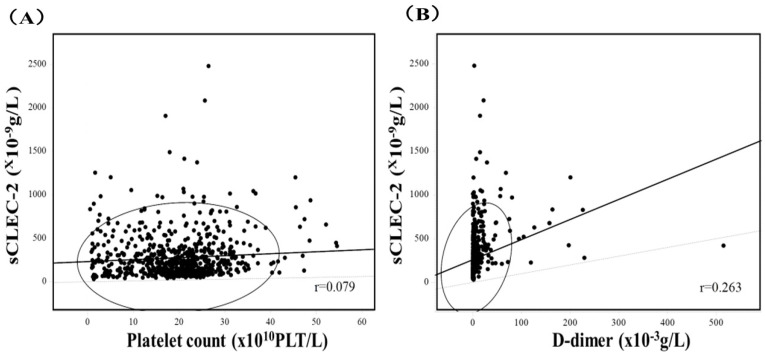
Relationships between sCLEC-2 level and platelet counts (**A**) and sCLEC-2 and D-dimer levels (**B**). (**A**) Y = 232 + 2.26 X (r = 0.079, *p* < 0.05); (**B**) Y = 260 + 2.30 X (r = 0.236, *p* < 0.001); sCLEC-2, soluble C-type lectin-like receptor 2.

**Figure 3 jcm-13-05980-f003:**
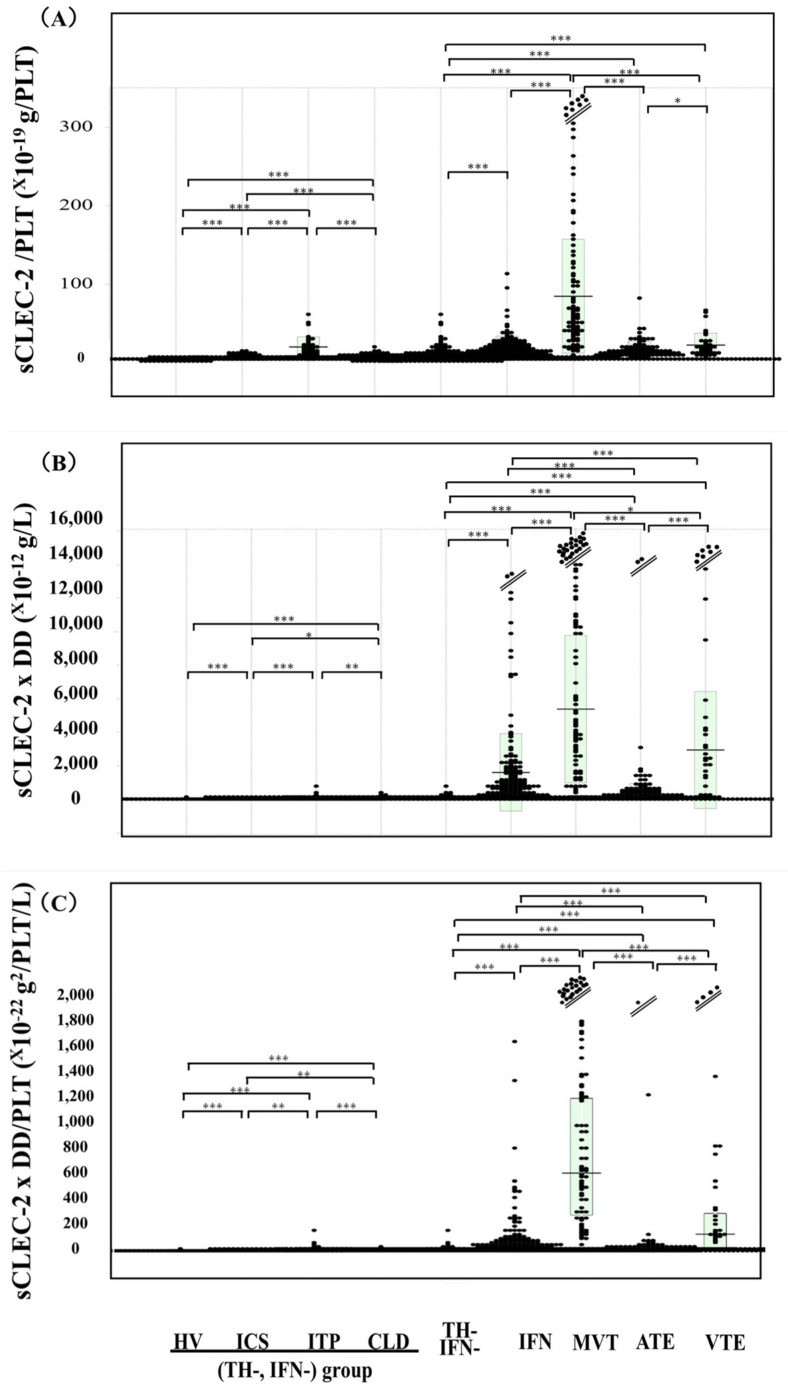
Plasma sCLEC-2/PLT (**A**), sCLEC-2 × D-dimer (**B**), and sCLEC-2 × D-dimer/platelet count (**C**) in HVs and patients with various diseases. sCLEC-2, soluble C-type lectin-like receptor 2; PLT, platelet count; HV, healthy volunteer; DD, D-dimer; ICS, indefinite compliant syndrome; ITP, immune thrombocytopenia; CLD, chronic liver disease; IFN, infection; TH-, without thrombosis; IFN-, without infection; MVT; microvascular thrombosis; VTE, venous thromboembolism; ATE, arterial thromboembolism; *, *p* < 0.05; **, *p* < 0.01; ***, *p* < 0.001.

**Figure 4 jcm-13-05980-f004:**
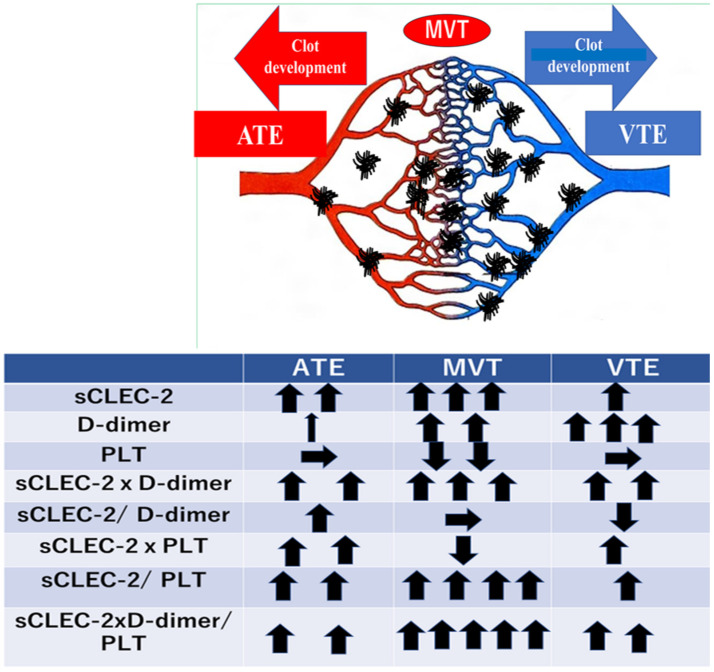
Relationships between thrombosis types and formulas with sCLEC-2 levels, D-dimer levels, and platelet counts. sCLEC-2, soluble C-type lectin-like receptor 2; PLT, platelet count; MVT; microvascular thrombosis; VTE, venous thromboembolism; ATE, arterial thromboembolism.

**Table 1 jcm-13-05980-t001:** Subjects.

	Sample	Patient	Age (Years Old)	Sex (F:M)	sCLEC-2 (×10^−9^ g/L)	Platelet Count (×10^10^ PLT/L)	D-Dimer (×10^−3^ g/L)
HV	79	79	21 (20–24)	39:40	92.2 (73.2–116)	21.6 (19.2–23.6)	0.20 (0.15–0.35)
ICS	52	52	57 (48–73)	27:25	145 (126–220) ***	21.0 (17.4–25.7)	0.41 (0.28–0.61) ***
ITP	35	22	50 (36–70)	15:7	77.0 (58.6–100) *	4.8 (4.1–7.3) ***	0.34 (0.24–0.52) **
CLD	177	177	66 (53–73)	90:87	113 (92.1–144) ***	20.7 (17.7–25.2)	0.43 (0.29–0.70) ***
TH-, IFN-	343	330	56 (31–71)	171:159	110 (85.8–139)	20.6 (17.0–24.4)	0.37 (0.24–0.56)
IFN	141	141	76 (58–84)	76:65	319 (233–477) ***	22.9 (16.3–28.2)	2.60 (1.20–5.05) ***
MVT	90	90	68 (52–79)	34:56	435 (278–675) ***	6.8 (2.6–12.4) ***	16.1 (6.00–30.8) ***
ATE	92	92	72 (65–78)	32:60	316 (234–388) ***	23.0 (18.8–27.3) *	0.95 (0.61–1.78) ***
VTE	36	22	69 (60–81)	16:6	315 (234–467) ***	18.9 (14.8–22.7) **	9.15 (2.95–18.4) ***

Data are expressed as median values (25th–75th percentile). sCLEC-2, soluble C-type lectin-like receptor 2; HV, healthy volunteer; ICS, indefinite compliant syndrome; ITP, immune thrombocytopenia; CLD, chronic liver disease; IFN, infection; TH-, without thrombosis; IFN-, without infection; MVT, microvascular thrombosis; VTE, venous thromboembolism; ATE, arterial thromboembolism; *, *p* < 0.05; **, *p* < 0.01; ***, *p* < 0.001 in comparison with HVs.

**Table 2 jcm-13-05980-t002:** ROC analyses on sCLEC-2 level, D-dimer level, and platelet count for their usefulness in the diagnosis of thrombosis.

	sCLEC-2 Level (×10^−9^ g/L)				D-Dimer Level (×10^−3^ g/L)				Platelet Count (×10^10^ PLT/L)			
	AUC	Sen	COF	OR	AUC	Sen	COF	OR	AUC	Sen	COF	OR
IFN vs. TH-, IFN-	0.973	90.8%	200	95.7	0.930	84.9%	0.8	4.35	0.579	58.8%	21.3	1.60
MVT vs. TH-, IFN-	0.977	93.0%	212	186	0.988	94.4%	1.1	17.0	0.836	81.1%	14.7	18.7
ATE vs. TH-, IFN-	0.979	91.5%	204	128	0.839	76.4%	0.6	3.23	0.628	60.2%	21.9	2.4
VTE vs. TH-, IFN-	0.967	91.7%	205	119	0.929	81.5%	0.7	4.88	0.563	58.1%	19.6	1.93
MVT vs. IFN	0.637	61.1%	371	2.50	0.850	77.8%	5.5	3.479	0.872	80.9%	14.5	18.1
ATE vs. IFN	0.520	50.5%	317	1.04	0.731	68.8%	1.5	2.13	0.535	50.8%	23.0	1.07
VTE vs. IFN	0.507	50.0%	321	1.01	0.712	74.5%	5.0	2.98	0.612	58.9%	20.3	1.23

sCLEC-2, soluble C-type lectin-like receptor 2; PLT, platelet count; HV, healthy volunteer; IFN, infection; TH-, without thrombosis; IFN-, without infection; MVT; microvascular thrombosis; VTE, venous thromboembolism; ATE, arterial thromboembolism; AUC, area under the curve; Sen, sensitivity; COF: cutoff; OR, odds ratio.

**Table 3 jcm-13-05980-t003:** sCLEC-2/PLT ratio, sCLEC-2xDD, and sCLEC-2xDD/PLT in various diseases.

	sCLEC-2/PLT (×10^−19^ g/PLT)	sCLEC-2xDD (×10^−12^ g/L)	sCLEC-2xDD/PLT (×10^−3^ g/L)
HV	4.30 (3.47–5.54)	18.9 (12.8–32.4)	0.94 (0.61–1.66)
ICS	7.82 (3.37–9.59) ***	71.3 (36.5–117) ***	3.54 (1.87–5.32) ***
ITP	17.2 (10.9–23.6) ***	28.5 (14.6–56.0)	7.06 (2.76–16.2) ***
CLD	5.29 (4.13–7.21) ***	49.1 (31.0–84.8) ***	2.28 (1.43–3.93) ***
TH-, IFN-	5.59 (4.20–8.03)	41.6 (22.6–74.6)	2.14 (1.13–4.08)
IFN	15.9 (11.1–23.4) ***	844 (335–1875) ***	42.3 (14.0–90.3) ***
MVT	63.1 (39.0–142) ***	6095 (2591–14,009) ***	943 (404–1821) ***
ATE	13.7 (10.8–19.1) ***	305 (172–620) ***	13.3 (7.79–27.9) ***
VTE	18.0 (12.2–26.0) ***	2600 (498–10,712) ***	141 (40.1–526) ***

sCLEC-2, soluble C-type lectin-like receptor 2; PLT, platelet count; HV, healthy volunteer; ICS, indefinite compliant syndrome; ITP, immune thrombocytopenia; CLD, chronic liver disease; IFN, infection; TH-, without thrombosis; IFN-, without infection; MVT; microvascular thrombosis; VTE, venous thromboembolism; ATE, arterial thromboembolism. ***, *p* < 0.001 in comparison with HVs.

**Table 4 jcm-13-05980-t004:** ROC analyses on sCLEC-2/PLT, sCLEC-2 × D-dimer, and sCLEC-2 × D-dimer/PLT for usefulness of diagnosis for thrombosis.

	sCLEC-2/PLT (×10^−19^ g/PLT)				sCLEC-2xDD				sCLEC-2xDD/PLT			
	AUC	Sen	COF	OR	AUC	Sen	COF	OR	AUC	Sen	COF	OR
IFN vs. TH-, IFN-	0.902	83.4%	9.72	25.7	0.981	94.3%	141	284	0.960	90.1%	7.43	72.8
MVT vs. TH-, IFN-	0.988	94.8%	18.2	352	0.998	96.7%	168	875	0.999	99.4%	54.5	11424
ATE vs. TH-, IFN-	0.899	83.7%	9.87	26.7	0.971	92.5%	134	156	0.934	87.0%	6.11	44.1
VTE vs. TH-, IFN-	0.930	86.1%	10.6	39.0	0.983	91.7%	131	124	0.965	87.5%	6.34	48.8
MVT vs. IFN	0.908	83.7%	27.5	27.2	0.839	77.8%	2000	12.4	0.941	85.3%	163	35.1
ATE vs. IFN	0.559	54.6%	14.5	1.46	0.717	66.7%	503	4.2	0.725	67.4%	21.5	4.27
VTE vs. IFN	0.557	54.6%	17.1	1.34	0.680	66.7%	1482	4.0	0.660	68.8%	74.8	4.85

sCLEC-2, soluble C-type lectin-like receptor 2; PLT, platelet count; DD, D-dimer; IFN, infection; TH-, without thrombosis; IFN-, without infection; MVT; microvascular thrombosis; VTE, venous thromboembolism; ATE, arterial thromboembolism; AUC, area under the curve; Sen, sensitivity; COF, cutoff; OR, odds ratio.

## Data Availability

The data presented in this study are available on request from the corresponding author. The data are not publicly available due to privacy restrictions.

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
