# Peer review of "Detection of Thrombosis Using Soluble C-Type Lectin-like Receptor-2 with D-Dimer Level and Platelet Count"

_jcm, 2024, doi:10.3390/jcm13195980_

Round 1
Reviewer 1 Report
Comments and Suggestions for Authors
In this manuscript the authors measured the plasma levels of sCLEC-2, along with plasma D-dimer level and platelet count, in healthy volunteers and patients with thrombosis, indefinite compliance syndrome, chronic liver disease, or infection. There are multiple major concerns in the study design and data analysis/presentation. Most critically, the data as presented does not support the conclusions.
The section of statistical analyses (page 3) should not be blank.
The data regarding the sensitivity and specificity of the monoclonal anti-sCLEC-2 for sCLEC-2 should be mentioned.
The authors equate elevated sCLEC-2 levels with platelet activation. However, there is no concrete data in the cited references to support this claim. Furthermore, the potential impact of age, sex and ethnicity on sCLEC-2 levels should be formally investigated. In conditions with elevated sCLEC-2, serial measurements are needed to help determine the optimal time to collect blood samples for the test.
The diagnosis groups are not clearly defined. For example, what is ICS and what was its diagnostic criteria? What was the status of various groups at the time of sample collection for the study? The elevation of D-dimer in the ITP group suggests some of the patients might had other compounding conditions. For the chronic liver disease group, how do the authors justify the grouping of patients with different etiologies or pathogenetic mechanisms? While a group of 73 patients with ACI without AF is mentioned in the Methods section, Table 1 lists an ATE group of 92 patients that is undefined. Similarly, the TH-IFN- group of 343 patients is not defined in the Methods section. What were the etiologies and severities of the infection group and the three thrombosis groups? The MVT group is also problematic, as it is quite heterogeneous in etiologies and pathogenetic mechanisms. Analyzing patients with different etiologies and pathogenetic mechanisms as one group invariably yields confusing and inconsistent results, as the result may be affected by the composition of the group. The lack of specifics for the various clinical groups limits the interpretability of the study results.
The authors mention in the legends of tables 2 and 4 that analysis of statistical significance was performed, yet present no statistical analysis data to support their conclusions.
Overall, the authors show that sCLEC-2, like D-dimer, is elevated not only in the thrombosis groups (MVT, ATE and VTE) but also in the other groups investigated such as ICS, ITP, CLD and infection. The claim that sCLEC-2 is specific for platelet activation should be supported by evidence. No statistical analysis is presented to show how the sCLEC-2 test is better than d-dimer for diagnosis of VTE, ATE and MVT, and the conclusion that “…….the formulars of sCLEC-2 with platelet count or D-dimer are useful for the diagnosis of thrombosis among patients with infection.”
The three figures are uninformative and superfluous.
Comments on the Quality of English LanguageThe English language requires extensive revision and editing.
Author Response
Thank you for your valuable comments. We have fully revised our manuscript in accordance with reviewer's comments.
Reviewer 1
In this manuscript the authors measured the plasma levels of sCLEC-2, along with plasma D-dimer level and platelet count, in healthy volunteers and patients with thrombosis, indefinite compliance syndrome, chronic liver disease, or infection. There are multiple major concerns in the study design and data analysis/presentation.
Comment 1
Most critically, the data as presented does not support the conclusions.
Response 1. Conclusions has been revised.
Comment 2.
The section of statistical analyses (page 3) should not be blank.
Response 2. The section of statistical analyses has been added.
Comment 3.
The data regarding the sensitivity and specificity of the monoclonal anti-sCLEC-2 for sCLEC-2 should be mentioned.
Response 3. This comment is not clear. The sensitivity (Limit of Quantification, LoQ) of the sCLEC-2 CLEIA method is 14.1 pg/mL (according to the package insert of the kit), allowing for the measurement of levels in samples from both healthy individuals and patients. The specificity of the antibodies used in the CLEIA reagents is described in detail in the analyses conducted by ELISA and Western blot methods in Papers A and B. These antibodies recognize both the 25 kD (shed form) and the 32, 40 kD (microparticle form) variants of sCLEC-2.
Comment 4.
The authors equate elevated sCLEC-2 levels with platelet activation. However, there is no concrete data in the cited references to support this claim.
Response 4. The relationship between elevated sCLEC-2 and platelet activation was already stated and cited between line 63-66.
Comment 5.
Furthermore, the potential impact of age, sex and ethnicity on sCLEC-2 levels should be formally investigated.
Response 5. This issue is studying and submitting by another group. In our preliminary examination, there were no significant difference in sCLEC-2 levels between different sex or young group and old group.
Comment 6.
In conditions with elevated sCLEC-2, serial measurements are needed to help determine the optimal time to collect blood samples for the test.
Response 6. It is reported that the measurement of sCLEC-2 is not affected by sampling time, sampling and measuring procedure. These papers were cited in this manuscript. Data was similar among serial measurements.
Comment 7.
The diagnosis groups are not clearly defined. For example, what is ICS and what was its diagnostic criteria?
Response 7. The definition of ICS has been added in Material section.
Comment 8.
What was the status of various groups at the time of sample collection for the study?
Response 8. Sampling time was stated in Method section.
Comment 9.
The elevation of D-dimer in the ITP group suggests some of the patients might had other compounding conditions.
Response 9. D-dimer 0.34 (0.24-0.52) in ITP is not markedly high and almost within normal range.
Comment 10.
For the chronic liver disease group, how do the authors justify the grouping of patients with different etiologies or pathogenetic mechanisms?
Response 10. Patients with liver cirrhosis, and hepatic cell carcinoma were excluded from this group. The CLD group was not associated with thrombosis or thrombocytopenia. This group was considered to non-thrombosis group without thrombocytopenia.
Comment 11.
While a group of 73 patients with ACI without AF is mentioned in the Methods section, Table 1 lists an ATE group of 92 patients that is undefined.
Response 11. Methods section has been revised. This study is to examine the difference of sCLECs formula with platelet count and D-dimer among non-thrombotic group, infectious group without thrombosis, MVT group, VTE group and ATE group.
Comment 12.
Similarly, the TH-IFN- group of 343 patients is not defined in the Methods section. What were the etiologies and severities of the infection group and the three thrombosis groups?
Response 12. Methods section has been revised. This study is to examine the difference of sCLECs formula with platelet count and D-dimer among non-thrombotic group, infectious group without thrombosis, MVT group, VTE group and ATE group.
Comment 13.
The MVT group is also problematic, as it is quite heterogeneous in etiologies and pathogenetic mechanisms. Analyzing patients with different etiologies and pathogenetic mechanisms as one group invariably yields confusing and inconsistent results, as the result may be affected by the composition of the group. The lack of specifics for the various clinical groups limits the interpretability of the study results.
Response 13. Methods section has been revised. This study is to examine the difference of sCLECs formula with platelet count and D-dimer among non-thrombotic group, infectious group without thrombosis, MVT group, VTE group and ATE group.
Comment 14.
The authors mention in the legends of tables 2 and 4 that analysis of statistical significance was performed, yet present no statistical analysis data to support their conclusions.
Response 14. Conclusions has been revised.
Comment 15.
Overall, the authors show that sCLEC-2, like D-dimer, is elevated not only in the thrombosis groups (MVT, ATE and VTE) but also in the other groups investigated such as ICS, ITP, CLD and infection. The claim that sCLEC-2 is specific for platelet activation should be supported by evidence. No statistical analysis is presented to show how the sCLEC-2 test is better than d-dimer for diagnosis of VTE, ATE and MVT, and the conclusion that “…….the formulars of sCLEC-2 with platelet count or D-dimer are useful for the diagnosis of thrombosis among patients with infection.”
Response 15. References which sCLEC-2 is specific for platelet activation, has been added. We did not compared sCLEC-2 to D-dimer. The conclusion has been revised as “the formulars of sCLEC-2 with platelet count or D-dimer are useful for the diagnosis of thrombosis among patients without infection.” and “the formulars of sCLEC-2 with platelet count or D-dimer are useful for the diagnosis of thrombosis using receiver operating characteristic (ROC) analyses for thrombosis group vs. non-thrombosis group without infection.”
Comment 16.
The three figures are uninformative and superfluous.
Response 16. All figures show the behavior of many parameters and have been revised. Summarized figure has been added.
Comment 17.
Comments on the Quality of English Language The English language requires extensive revision and editing.
Response 17. Revised manuscript has been reviewed by MDPI English editing service.
Reviewer 2 Report
Comments and Suggestions for Authors
The authors performed a study in which they used sCLEC-2, sCLEC-2/platelet count (sCLEC-2/PLT), sCLEC-2xD-dimer (eCLEC-2xDD) and sCLEC-2xDD/PLT in order to detect thrombotic disorders.
They reported that sCLEC-2 is useful for the diagnosis of thrombosis, and forms of sCLEC-2 with platelet count or D-dimer are useful for the diagnosis of thrombosis among patients with infection.
The topic of the chosen study is interesting.
Some suggestions for authors would be:
1. Abstract: - in my opinion, the purpose of the study is presented more in the methods.
2. Introduction: - I think it should start with some general data about thrombosis.
3.Line 225- it is written mbosis. It must be change with thrombosis
Author Response
Thank you for your kind and valuable comments. We have fully revised ourmanuscript in accordance with reviewer's comments.
Reviewer 2
The authors performed a study in which they used sCLEC-2, sCLEC-2/platelet count (sCLEC-2/PLT), sCLEC-2xD-dimer (eCLEC-2xDD) and sCLEC-2xDD/PLT in order to detect thrombotic disorders.
They reported that sCLEC-2 is useful for the diagnosis of thrombosis, and forms of sCLEC-2 with platelet count or D-dimer are useful for the diagnosis of thrombosis among patients with infection.
The topic of the chosen study is interesting.
Some suggestions for authors would be:
Comment 1.
- Abstract: - in my opinion, the purpose of the study is presented more in the methods.
Response 1. The purpose has been added in method section.
Comment 2.
Introduction: - I think it should start with some general data about thrombosis.
Response 2. Introduction has been revised.
Comment 3.
Line 225- it is written mbosis. It must be change with thrombosis
Response 3. “mbosis” has been changed to “thrombosis”.